# A Review of Colonoscopy in Intestinal Diseases

**DOI:** 10.3390/diagnostics13071262

**Published:** 2023-03-27

**Authors:** Seung Min Hong, Dong Hoon Baek

**Affiliations:** Department of Internal Medicine, Pusan National University School of Medicine, Biomedical Research Institute, Pusan National University Hospital, Busan 49421, Republic of Korea

**Keywords:** colonoscopy, intestinal diseases, review

## Abstract

Since the development of the fiberoptic colonoscope in the late 1960s, colonoscopy has been a useful tool to diagnose and treat various intestinal diseases. This article reviews the clinical use of colonoscopy for various intestinal diseases based on present and future perspectives. Intestinal diseases include infectious diseases, inflammatory bowel disease (IBD), neoplasms, functional bowel disorders, and others. In cases of infectious diseases, colonoscopy is helpful in making the differential diagnosis, revealing endoscopic gross findings, and obtaining the specimens for pathology. Additionally, colonoscopy provides clues for distinguishing between infectious disease and IBD, and aids in the post-treatment monitoring of IBD. Colonoscopy is essential for the diagnosis of neoplasms that are diagnosed through only pathological confirmation. At present, malignant tumors are commonly being treated using endoscopy because of the advancement of endoscopic resection procedures. Moreover, the characteristics of tumors can be described in more detail by image-enhanced endoscopy and magnifying endoscopy. Colonoscopy can be helpful for the endoscopic decompression of colonic volvulus in large bowel obstruction, balloon dilatation as a treatment for benign stricture, and colon stenting as a treatment for malignant obstruction. In the diagnosis of functional bowel disorder, colonoscopy is used to investigate other organic causes of the symptom.

## 1. Introduction

Colonoscopy is an examination of the colorectum and terminal ileum undertaken by inserting a scope with a camera device and flexible light source through the anus. Since colonoscopy was first performed in the 1960s [1], it has been used as a key diagnostic and therapeutic tool for various intestinal diseases. There are many types of intestinal diseases, and they can be classified into infectious disease, inflammatory bowel disease (IBD), neoplasm, functional bowel disorder, bleeding, and others. Colonoscopy can visualize lesions associated with these diseases and find inflammation, ulcers, neoplasms, and hemorrhages. In addition, it provides information on macroscopic findings and enables tissue sampling by inserting instruments through various channels [2]. Moreover, because of the development of endoscopic resection techniques such as endoscopic submucosal dissection (ESD) and endoscopic mucosal resection (EMR), endoscopic resection is used as the main treatment for early colorectal cancer [3]. Colonoscopy also plays an important role in large bowel obstruction (LBO). Colonoscopy not only enables the diagnosis of various diseases of LBO, but it is also useful as a treatment for balloon dilatation in benign stricture and metal stent insertion in malignant obstruction [4]. Additionally, when gastrointestinal bleeding occurs, endoscopic hemostasis is performed through endoclipping or an electronic surgical unit, and endoscopic perforation treatment can also be used for bowel perforation. However, in functional bowel disorders, colonoscopy is used to exclude other organic causes rather than to diagnose the disease itself [5]. As such, colonoscopy is widely used in various diseases and clinical situations. Herein, we summarize the use and role of colonoscopy in various intestinal diseases.

## 2. Colonoscopy in Intestinal Diseases

### 2.1. Infectious Diseases

An intestinal tract infection can cause abdominal pain, fever, diarrhea, loose stool, and bloody or mucoid stool, and is caused by bacteria, viruses, or parasites. Common causes of infectious enterocolitis include *Yersinia enterocolitica*, *Salmonella*, *Shigella*, *Escherichia coli*, *Campylobacter, Clostridium difficile*, *Mycobacterium tuberculosis,* cytomegalovirus (CMV), and *Entamoeba histolytica* [6]. In such infectious intestinal diseases, colonoscopy is more useful for diagnostics than therapeutics. In most cases of infectious colitis, endoscopic findings are accompanied by edema, redness, ulceration, exudation, and mucosal friability [7]. Therefore, it is difficult to discriminate between the causative microorganisms that cause infection using only endoscopic findings. Yet, the location of the lesion can be an important clue when making a differential diagnosis. Table 1 summarizes the types of infectious enterocolitis that predominate according to the location of the lesion. Especially in immunocompromised people or men who have sex with men, infectious diseases such as *Neisseria gonorrhea, Chlamydia trachomatis*, herpes simplex virus, human papilloma virus, syphilis, and *Treponema pallidum* can occur in the rectum. In these conditions, symptoms such as anorectal pain, tenesmus, and mucopurulent discharge may be present [8].

Although most cases of infectious enterocolitis yield similar endoscopic macroscopic findings, some cases of infectious enterocolitis have characteristic endoscopic findings. *Yersinia* enterocolitis is caused by infection with *Yersinia enterocolitica*, a Gram-negative bacillus distributed worldwide. *Yersinia* enterocolitis usually affects the terminal ileum or right colon, but occasionally the left colon. Because the right colon and terminal ileum are frequently involved, full colonoscopy should be considered to confirm *Yersinia* infection [9]. Rutgeerts et al. reported that *Yersinia* enteritis in the terminal ileum is characterized by large ulcers in the form of granular mucosa [10]. Arai et al. also reported multiple granular elevated lesions in *Yersinia* ileitis involving the terminal ileum [11]. *Yersinia* enterocolitis yields inflammatory findings accompanied by granular mucosa of the distal ileum, and is often mistaken for Crohn’s disease (CD) because of its location [12,13,14,15]. Therefore, diagnosis of *Yersinia* enterocolitis should not be made simply by endoscopic findings; other clinical features and clinical findings derived through laboratory tests such as stool tests should be comprehensively considered.

Gastrointestinal (GI) salmonellosis is a disease caused by infection of the GI tract with *Salmonella* species. *Salmonella* mainly affects the distal ileum and the right colon, but in some cases the entire colon may be involved; thus, full colonoscopy should be considered when *Salmonella* infection is suspected, such as *Yersinia* enterocolitis [16]. It is difficult to differentiate *Salmonella* enterocolitis only by endoscopic findings because it yields non-specific acute inflammatory findings, such as mucosal redness, mucosal friability, ulcers, and erosion [17,18]. In severe *Salmonella* enterocolitis involving the whole colon, care must be taken not to confuse it with ulcerative colitis (UC). Moreover, care should be taken not to confuse it with CD when the right colon is severely involved [16].

Shigellosis presents with fever and watery diarrhea, progressing to invasive, hemorrhagic colitis [19]. Upon endoscopy, shigellosis shows mucosal redness, punctate spots, mucosal edema, irregular ulcers, mucosal friability, and exudate [20]. Sometimes in severe shigellosis, the ulcers coalesce and form a circular shape [21]. Although shigellosis mainly affects the left colon, particularly the rectosigmoid colon, it can extend to the proximal part beyond the rectosigmoid colon, and it may present as pancolitis in 15% of cases [20,22]. Shigella can be confused with UC because it shows ulceration endoscopically with diarrhea and bleeding, and the involved area is similar to that in UC.

Enterohemorrhagic *E. coli* enterocolitis (EHEC) can cause hemorrhagic colitis, diarrhea, and hemolytic uremic syndrome [21]. Several studies have reported that inflammation may appear in the entire colorectum, but is more prevalent in the right colon [23,24,25,26]. When severe inflammation occurs, marked swelling, hemorrhage, and dark red erythema may appear in the right colon, which may be similar to the endoscopic findings of ischemic colitis. Moreover, ischemic colitis and EHEC have similar histological findings [27,28,29]. However, they can be differentiated by their common location of involvement. Ischemic colitis usually occurs in the left colon, especially in the watershed area, whereas EHEC enterocolitis occurs more severely in the right colon [21,30].

Pseudomembranous colitis (PMC) is characterized by the presence of numerous yellowish-white plaques forming a pseudomembrane on the colonic mucosa. Endoscopic findings are characterized by multiple yellowish or creamy mucosal plaques [31]. The most common cause of PMC is *Clostridium difficile* [32]. However, it can also be rarely caused by *Clostridium ramosum*, *Entamoeba histolytica, E. coli* O157:H7, *Klebsiella oxytoca*, *Salmonella* species, *Shigella* species, CMV, chemical agents and medications, IBD, and ischemic colitis [33]. *C. difficile*-associated PMC is caused by *C. difficile* toxins, and the use of antibiotics is the greatest risk factor for *C. difficile* overgrowth. PMC usually involves the left colon, but may involve the entire colon in up to approximately one-third of cases [19,21,34]. However, colonoscopy does not always show typical positive findings in pseudomembranous colitis. Bergstein et al. reported that 16 of 29 (55%) patients with confirmed *C. difficile* had endoscopic confirmation of pseudomembrane, and non-specific colitis was found in 4 (14%) [35]. Additionally, Gebhard et al. reported that in the early course of *C. Difficile*-associated PMC, tiny round yellowish spots, different from the usual findings of extensive PMC, could be seen [36]. Colonoscopy can also be used for therapeutic purposes in *C. difficile* infection. Fecal microbiota transplantation for the treatment of refractory *C. difficile* infection, or for the prevention of recurrence, can be administered via colonoscopy [37].

To diagnose intestinal tuberculosis, tissue sampling is required, so colonoscopy is essential [38,39]. Since intestinal tuberculosis often invades the terminal ileum, the terminal ileum should be observed when performing colonoscopy [40]. Endoscopic findings of intestinal tuberculosis include erosions, aphthous ulcers, circumferential ulcers, round- or irregular-shaped ulcers with circumferential arrangements, multiple nodules, ileocecal deformity, and luminal narrowing [39,41]. Since intestinal tuberculosis tends to involve the ileocecal area and the endoscopic findings are similar to those of CD, care must be taken in making the differential diagnosis. Intestinal tuberculosis more frequently shows a patulous ileocecal valve, scars, and pseudopolyps, and it tends to involve fewer than four segments [42]. Although tissue collection is essential for the diagnosis of intestinal tuberculosis, the probability of confirming intestinal tuberculosis via pathological findings using a biopsy tissue or culture is only 38.7% [43]. Although the confirmation rate via tissue sampling is low, it is also important to confirm the endoscopic findings for the sake of diagnosis.

CMV disease is caused by the reactivation of a latent virus, and is mainly seen in immunocompromised individuals, such as organ transplant recipients [21,44,45]. The GI tract is one of the common organs involved in CMV disease [46]. The diagnostic gold standard for GI CMV disease is the presence of CMV in a tissue sample. However, there may be sampling error and the diagnostic yield is low, so it is not always possible to obtain meaningful results for diagnosis [47,48]. An important endoscopic finding of GI CMV disease is a well-defined ulcer with a punch-out appearance. Occasionally, endoscopic findings may show nonspecific erosions, ulcers, hemorrhagic spots, and granularity and friable mucosa that are difficult to distinguish from UC [49,50,51].

Amoebic colitis is caused by intestinal infection with *Entamoeba histolytica*. Amoebiasis does not cause symptoms in most cases, but approximately 10% of infected people develop symptoms [52]. Colonoscopy can be a good tool for diagnosing amebic colitis. In particular, the microscopic confirmation of trophozoites that phagocytize red blood cells by performing an endoscopic biopsy sample is the most reliable method for diagnosing amebiasis [53]. Endoscopically, amoebic colitis is frequently identified in the cecum or ascending colonm and appears mainly as an ulcerative lesion. The size of the lesion varies from several millimeters to several centimeters, and it shows a clear border with the surrounding normal mucosa and is covered with exudate. In the early stages of the disease, only inflammatory findings, such as mucosal redness, may be seen [53,54]. Tissue biopsy is not diagnostic two-thirds of cases [55,56].

### 2.2. Inflammatory Bowel Diseases

IBD is classified into CD and UC. Until the 1990s, the treatment goal for IBD was mainly clinical remission. However, as the treatment paradigm has recently changed, the role of endoscopy is becoming more important. An Update on the Selecting Therapeutic Targets in Inflammatory Bowel Disease (STRIDE-II) published in 2021 suggested endoscopic healing as a long-term target along with normalized quality of life [57]. Endoscopy, especially ileocolonoscopy, is an essential tool for diagnosing IBD, confirming disease activity, assessing treatment effects, performing colorectal cancer screening, and providing treatment such as endoscopic dilatation [58,59,60,61,62,63,64]. UC and CD show differences in endoscopic findings, and they are very helpful in diagnosis. CD mainly shows segmental involvement, aphthous ulcers, serpentious, longitudinal ulcers, large deep ulcers, rectal sparing, anal or perianal disease, and a cobble stone appearance. Conversely, UC shows a continuous lesion, loss of vascular pattern, granular mucosa, erosion, and rectal involvement [65,66]. Generally, CD can involve the entire GI tract, and UC affects only the colorectum. However, inflammation of the terminal ileum, i.e., backwash ileitis, is found in 10% of patients with diffuse active UC [67]. Since CD often invades the terminal ileum, it is essential to observe the terminal ileum during colonoscopy [66]. Histopathological evaluation through colonoscopic biopsy, especially the identification of granuloma specific to CD, helps to differentiate IBD [68]. However, not all tissue samples of CD show granuloma on histopathological examination. The rate of confirmation of granuloma through endoscopic biopsy in CD is as low as 15% to 36% [66].

Mucosal healing is a strong predictor of an IBD patient’s long-term outcome [69,70]. In UC, mucosal healing leads to clinical remission and reduces the risk of colon cancer. In CD, mucosal healing reduces surgery and hospitalization rates [71,72]. Table 2 summarizes the endoscopic scoring system commonly used in IBD. Endoscopic evaluation is required to evaluate mucosal healing. Since UC occurs only in the colorectum, colonoscopy is essential to evaluate disease activity. Endoscopic severity assessment scoring systems used for UC include the Mayo endoscopic subscore (MES), Ulcerative Colitis Endoscopic Index of Severity (UCEIS), and Ulcerative Colitis Colonoscopic Index of Severity (UCCIS). The MES is a part of the Mayo score and is widely used in clinical practice. The MES classifies UC into normal or inactive disease, mild disease (erythema, decreased vascularity, mild friability), moderated disease (marked erythema, absent vascularity, friability, and erosions), and severe disease (spontaneous bleeding and ulceration) [73]. UCEIS is a scoring system that evaluates each of the nine items of vascular pattern, mucosal erythema, mucosal surface, mucosal edema, mucopus, bleeding, incidental friability, contact friability, erosions and ulcers, and extent of erosions or ulcers [74]. UCCIS uses four parameters: granularity, vascular pattern, ulceration, and bleeding/friability [75]. The first endoscopic scoring system for CD was the Crohn’s Disease Endoscopic Index of Severity (CDEIS), but it is difficult to use in clinical practice because of its complexity. The subsequent Simple Endoscopic Score for Crohn’s Disease assesses the degree of ulceration, ulcerated surface, inflamed surface, and stenosis for five defined bowel segments (the rectum, sigmoid and descending colon, transverse colon, ascending colon, and terminal ileum) to classify the disease activity [76].

In IBD, colonoscopy is required for the screening of cancer and dysplasia along with evaluating disease activity. Chronic inflammation of the intestine due to IBD increases the risk of colorectal cancer. The incidence of colorectal cancer in patients with IBD is two- to three-fold [77,78]. The increased risk of colorectal cancer in patients with inflammatory bowel disease can be managed through periodic colonoscopy surveillance. The 2019 BSG and ECCO guidelines recommend starting surveillance 8 years after symptom onset in patients with inflammatory bowel disease with colon involvement. However, if PSC is present, it is recommended to start monitoring at the time of diagnosis [79,80]. In addition, the surveillance interval is classified according to the severity of the disease and is recommended at intervals of 1–5 years. When determining the surveillance interval, the presence of primary sclerosing cholangitis, severity of inflammation, family history, dense pseudopolyps, and dysplasia should be considered [81]. A Cochrane Database systemic review and meta-analysis reported that the surveillance colonoscopy group of patients with IBD showed lower cancer detection (3.2% vs. 1.8%, odds ratio (OR), 0.58; 95% confidence interval (CI), 0.42–0.80, *p* < 0.001), lower colorectal cancer-related mortality (22.3% vs. 8.5%, OR, 0.36; 95% CI, 0.19–0.69, *p* = 0.002), and a higher rate of early-stage colorectal cancer (7.7% vs. 15.5%, OR, 5.40; 95% CI, 1.51–19.30, *p* = 0.009) than the no-surveillance group [82]. Continuous surveillance colonoscopy is required to reduce the increased risk of colorectal cancer in patients with IBD.

The surveillance of dysplasia should also be performed in inflammatory bowel disease. The 2019 BSG and ECCO guidelines recommend using high-definition endoscopy rather than standard-definition, and chromoendoscopy rather than white light endoscopy. For chromoendoscopy, the use of methylene blue or indigo carmine is recommended [79,80]. However, chromoendoscopy can be considered impractical for practitioners because it takes a long time and requires several preparations. Therefore, instead of chromoendoscopy, high-definition endoscopy can also be used as a good alternative [83,84]. Previously, biopsies were performed four times every 10 cm when conducting surveillance colonoscopy in patients with IBD, but their effectiveness is controversial. Random biopsy is considered to be problematic because of the low dysplasia detection rate and prolonged procedure time. Moreover, the 2019 BSG and ECCO guidelines recommend target biopsy instead of random biopsy. Therefore, random biopsy can be considered in selected cases [79,80,85]. In past guidelines, surgical proctocolectomy was recommended when dysplasia was identified on surveillance colonoscopy for IBD. However, the recent trend is to attempt endoscopic resection according to the lesion characteristics [83]. Surgical operation is considered for non-visible dysplasia confirmed by random biopsy [86]. On the other hand, macroscopically identified dysplasia lesions can be removed endoscopically. Endoscopic resection should be performed by a skilled therapeutic endoscopist, and it is determined depending on the shape, size, site, and submucosal invasion of the lesion [86]. Endoscopic resection methods include EMR, ESD, modified EMR (mEMR), and hybrid ESD. It is recommended to perform endoscopic resection when in the endoscopic remission state [81].

IBD is accompanied by various bowel complications, the most representative of which is stricture. Stricture occurs primarily in patients with CD and occurs in up to 33% of patients with CD 10 years after diagnosis [87]. If symptoms occur due to stricture, surgical or endoscopic treatment is required. Since repetitive surgical operations can lead to short bowel syndrome, endoscopic balloon dilatation can replace surgery to preserve the bowel. Endoscopic balloon dilatation should be avoided in patients with fistulas, deep ulcers, or long strictures >5 cm [88,89]. In one study, the technical success rate of endoscopic balloon dilatation was 89% and the clinical success rate was 81% [90]. However, repeated endoscopic dilation is often required because of the high recurrence rate of stricture. Gustavsson et al. performed 776 dilatations in 178 patients with CD. At the 5-year follow-up, only 52% of the patients did not require additional dilatation or only needed one additional dilatation. Complications occurred in 5.3% of patients, and 36% underwent surgery [91]. Ferlitsch et al. reported that after endoscopic dilatation for CD stricture, repeated dilatation was performed in 31% of cases, and surgical resection was performed in 28% [92]. If the length of the stricture is short (<4 cm), stricture of the surgical anastomosis is the most suitable target for balloon dilatation. However, surgical treatment should be considered in cases of multiple stenosis, >4–5 cm, fistula, or abscess [90]. Although strictures are more commonly observed in CD than in UC, if strictures are found in UC patients with a long morbidity period, a biopsy is necessary because there is a risk of dysplasia or colorectal cancer. The characteristics of malignant stricture in UC are as follows: first, it occurs 10–20 years after the onset of UC; second, it is more common in the proximal than in the splenic curve; and third, it is often expressed as a symptom of colonic obstruction [93].

### 2.3. Neoplasms

Colorectal cancer (CRC) is a major cause of morbidity and mortality throughout the world. It accounts for over 10% of all cancer incidence [94]. It is the third most common cancer worldwide and the second most common cause of death [94]. Most guidelines, including those from the American Cancer Society [95], the US Preventive Services Task Force [96], and the European Society of Gastrointestinal Endoscopy (ESGE) [97], recommend screening for CRC in average-risk individuals beginning at the age of 45 or 50 years. Both colonoscopy and sigmoidoscopy can detect and remove polyps, potentially preventing malignant transformation and decreasing CRC mortality and incidence. To date, four large randomized controlled trials comparing flexible sigmoidoscopy screening with no screening showed reductions in CRC incidence (18–23%) and CRC mortality (22–33%) [98,99,100,101]. These findings provide substantial protection against CRC diagnosis and death, and the benefits can last for up to 17 years [102]. Randomized controlled trials of screening colonoscopy are ongoing, but definitive results will not be available until 2022 or 2026–2027 [103,104,105]. Cohort and case–control studies found an association between lower endoscopy and reduced CRC mortality and incidence. A large prospective cohort study of nearly 89,000 nurses and other healthcare professionals found that, over 24 years of follow-up, colonoscopy was associated with a 68% reduction (95% CI, 0.55–0.76) in CRC-specific mortality compared with no exposure to colonoscopy [106]. Individuals who underwent colonoscopy with polypectomy were found to have a 43% reduction in CRC incidence compared to those with no lower endoscopy [106]. However, cohort studies probably overestimate the real-world effectiveness of colonoscopy because of the inability to adjust for important factors such as incomplete adherence to testing and the tendency of healthier persons to seek preventive care. In a Canadian case–control study, any colonoscopy was associated with a 37% reduction in the odds of CRC death [107]. Similar case–control studies using the Surveillance, Epidemiology, and End Results (SEER)-Medicare and Veterans Administration data also found approximately 60% reductions in CRC death associated with colonoscopy, with similar differences by site [108,109]. However, these three case–control studies were unable to determine indications for colonoscopy, and excluded colonoscopies performed within 6 months of CRC diagnosis, likely introducing bias. In a meta-analysis conducted with 13 cohorts including 4,713,778 individuals and 16 case–control studies, colonoscopy screening not only reduced the incidence of colorectal cancer by 52% (risk ratio (RR): 0.48, 95% CI, 0.46–0.49), but also reduced colorectal cancer related mortality by 62% (RR: 0.38, 95% CI, 0.36–0.40) [110]. Flexible sigmoidoscopy and colonoscopy are both recommended CRC screening strategies, but their relative effectiveness is unclear. According to the case–control study using the SEER-Medicare database, screening colonoscopy was associated with a greater reduction of 74% (OR 0.26, 95% CI, 0.23–0.30) in CRC mortality compared to screening sigmoidoscopy, which was associated with a 35% reduction (OR 0.65, 95% CI, 0.48–0.89) in CRC mortality. Additionally, screening colonoscopy was found to be more effective in reducing mortality in the distal colon compared to the proximal colon.

Improving colonoscopy screening results is crucial for the early detection and prevention of colorectal cancer [111]. Recording quality indicators is essential for assessing the effectiveness of population-based colonoscopy screening programs. The quality indicators vary between countries, such as the United States [111] and the United Kingdom [112], but they generally include the following:Consent obtained—Ensuring informed consent is obtained from patients before the procedure;Cecal insertion rate—A high rate (97% or higher in the US, 90% minimum in the UK) indicates successful navigation of the colonoscope to the cecum;Adequate bowel preparation—A clean colon is necessary for accurate visualization; the suggested rates are 85% or higher in the US and 90–95% in the UK;Adenoma detection rate (ADR)—The percentage of patients with at least one adenoma detected during colonoscopy. Higher rates (25% or more in the US, 35–40% in the UK) indicate better screening quality;Withdrawal time—Time taken for the colonoscope to be withdrawn after reaching the cecum. Longer times (6 min or more in the US, 6–10 min in the UK) are associated with improved adenoma detection;Complication rates—Low rates of complications, such as perforation (1/1000 or less) and bleeding after polypectomy (1% or less in the US, 1/100 or less in the UK);Polyp retrieval rate—The percentage of removed polyps that are successfully retrieved for histopathological examination (90–95% in the UK).

The NordICC (Nordic-European Initiative on Colorectal Cancer) study highlights the importance of quality control in population-based colonoscopy screening programs. A significant issue identified in this study is the low quality of colonoscopy screenings, which can affect the effectiveness of these programs in detecting and preventing CRC [113]. The ongoing NordICC study aims to evaluate the long-term performance of colonoscopy screening and the impact of quality control measures. In the next 5 years, the study is expected to yield valuable insights into the effectiveness of various quality indicators in improving colonoscopy screening results [104]. By examining these results, healthcare professionals and policymakers can make informed decisions about implementing and refining population-based colonoscopy screening programs.

Colonoscopy describes the size and shape of neoplasms found during the diagnostic process, and also can estimate the tumor’s malignant potential and invasion depth. Generally, the macroscopic appearance of colonic lesions is described using the Paris classification. According to the Paris classification, among neoplastic lesions with superficial morphology, those taller than the height of the biopsy forcep (2.5 mm) are defined as polypoid, and those that are not are defined as non-polypoid. Polypoid lesions are classified as I, which are further classified as pedunculated (lp), sessile (ls), and semi-pedunculated (Isp). Nonpolypoids are classified as slightly elevated (lla), flat (llb), slightly depressed (llc), and excavated (lll) [114]. Classifying the endoscopic macroscopic type helps to understand the characteristics of the lesion and select an appropriate endoscopic resection method for the lesion. Macroscopic findings are important clues to determine the invasion depth of the lesion. Currently, the indication for endoscopic treatment of colorectal cancer is early colorectal cancer that invades the mucosa or submucosa to <1000 μm [115,116,117]. Therefore, it is necessary to accurately determine the invasion depth of the lesion before endoscopic resection to avoid unnecessary procedures. Representative morphological features suggesting submucosal invasion of colorectal cancer include loss of lobulation, demarcated depression area, stalk swelling, excavation, fullness, ulcer bleeding, fold convergency, and non-lifting signs [118]. Kudo et al. first proposed a classification method called pit pattern using a magnifying endoscopy and indigo carmine dye [119]. This classification classifies into five types, from type I to type V, and according to each classification, the tissue type and invasion depth of the colorectal tumor can be estimated. As electronic chromoendoscopy can replace chromoagents, the Tumor Narrow Band Imaging (NBI) Interest Group in 2010 proposed the NBI International Colorectal Endoscopic (NICE) classification, which is a method of classifying colorectal lesions according to NBI findings without magnifying endoscopy [120]. Additionally, in 2014, the Japan NBI Expert Team (JNET) proposed the JNET classification using magnifying endoscopy and NBI. The JNET classification divides colorectal lesions into four types (types 1, 2A, 2B, and 3) using surface and vascular patterns, and this is different from the NICE classification, in that the JNET classification can distinguish between benign lesion and mucosal cancer [121].

A subepithelial lesion (SEL) is also a neoplastic lesion that is frequently encountered in clinical practice. SELs can occur in any segment of the colon. As the rate of screening colonoscopy increases, the number of SEL cases is also increasing [122]. SELs can be benign and malignant, so making an accurate diagnosis is very important. For the diagnosis of SELs, first, it is important to confirm the macroscopic findings of the lesion through colonoscopy. Most SELs are lesions < 20 mm, covered by normal mucosa. The color of the surface mucosa varies from normal pinkish to yellowish, bluish, whitish, and reddish. The consistency of the lesion can be assessed by touching it with biopsy forceps. If the cushion sign is positive, it is often a lipoma or lymphangioma. In addition, when pulsation is observed in the lesion, it can be considered as a blood vessel. Rapid growth in size or surface ulceration can be considered as findings suggesting malignancy [123]. An SEL may not be an intraluminal lesion, and may instead be compression caused by an external structure. If the location and pattern of the lesion change through air control or posture change, the possibility of extraluminal compression should be considered. A prospective study reported that when 100 SELs were evaluated, endoscopic identification of the intramural or extramural location of the lesion showed a sensitivity of 98% and a specificity of 64%. This finding suggests that extramural lesions may be mistaken for intramural lesions by endoscopy alone [124]. Therefore, when extramural compression is suspected, performing additional modalities such as endoscopic ultrasonography (EUS) and computed tomography (CT) is helpful for diagnosis. When using EUS, the accuracy of distinguishing between extramural and intramural lesions reaches approximately 90% [124].

EUS is useful for the differentiation of intramural SELs by evaluating the originating layer and echogenicity of the SEL. Table 3 [125,126] summarizes the layers and echogenicity of representative colonic SELs commonly encountered. In addition to layer and echogenecity, there are clues that are helpful for diagnosis during EUS. First, when there is erosion or ulceration on the surface of the SEL, it is likely a malignancy, such as submucosal tumor like-cancer, metastatic cancer, a neuroendocrine tumor, lymphoma, or a GI stromal tumor. A lipoma has a yellow surface with a positive cushion sign, and when a biopsy is performed, a characteristic naked fat sign is observed. Lymphangioma also has a positive cushion sign, and unlike a lipoma, it has a pale, transparent surface. In addition, a lymphangioma is characterized by anechoic cystic spaces with septations when EUS is performed. It is impossible to completely discriminate all SELs with only EUS. In one study, the concordance between EUS and a histopathologic diagnosis was 79.3% [127]. In summary, to diagnose SEL, the location of the lesion, macroscopic findings, and EUS findings should be comprehensively considered, and if necessary, additional imaging modalities such as CT and magnetic resonance imaging should be used [126].

Polypectomy, EMR, and ESD are representative endoscopic resections for lower GI neoplasms. Polypectomy includes cold snare polypectomy and hot snare polypectomy. At present, mEMR and hybrid ESD are also widely used. mEMR includes EMR after circumferential precutting, EMR with cap, anchored snare-tip EMR, EMR with band ligation, and EMR using a dual-channel endoscope. mEMR overcomes the limitations of polypectomy and conventional EMR by capturing and excising the deep submucosa [128]. The type of endoscopic resection should be selected differently according to the size and shape of the lesion and invasion depth. The United States Multi-Society Task Force (USMSTF) recommends cold snare polypectomy for ≤9 mm non-pedunculated polyps. After performing endoscopic imaging assessment for polyps ≥ 10 mm, polypectomy or EMR is recommended for non-invasive lesions, and EMR or ESD is recommended for lesions with suspected minimal or moderate risk of submucosal invasion. Hot snare polypectomy is recommended for pedunculated lesions, and prophylactic ligation of the stalk is recommended for head size ≥ 20 mm or stalk thickness ≥ 5 mm [129]. The ESGE recommends almost the same guidelines as the USMSTF, except that a stalk width ≥ 10 mm is the criterion for prophylactic hemostasis in pedunculated lesions [130].

### 2.4. Large Bowel Obstruction

Nearly one-quarter of bowel obstructions occur in LBO [131]. Since the tension of the large bowel wall follows the LaPlace law, the greatest increase in tension in the cecum, which has the widest diameter in the large intestine, increases the risk of ischemia and perforation if the diameter increases by >12 cm. The causes of LBO are diverse, with colorectal cancer being the most common (60%), followed by colonic volvulus (10–15%) and diverticulitis (5–10%) [132]. The remaining LBOs are caused by conditions such as anastomotic strictures, IBD, hernia, intussusception, adhesion, endometriosis, and functional disorders of the colon.

The typical site of colonic volvulus is within 25 cm of the anal verge. The coexistence of sigmoid volvulus with chronic immobility, chronic constipation, laxative uses, and neuropsychiatric disorders is frequently emphasized [133]. Emergency endoscopic decompression is helpful for diagnosing and treating sigmoid colon volvulus, and the success rate of endoscopic decompression has been reported to be approximately 70–80% [134]. However, contraindications to endoscopic decompression include perforation peritonitis, suspicion of bowel gangrene manifesting as features of sepsis, and persistent hematochezia. Immediate resuscitation and surgery are recommended in these cases. Endoscopic decompression is not a definitive treatment option in most patients, and high recurrence rates (30–90%) are reported. Therefore, a laparoscopic approach (bowel resection) within 2–5 days following endoscopic decompression is recommended [135]. Benign colonic strictures are mainly treated with balloon dilation, and malignant colonic strictures are mainly treated with colonoscopic stent implantation. Indications for balloon dilatation include surgical anastomotic stricture, stricture due to IBD (mainly CD), stricture caused by non-steroidal anti-inflammatory drug (NSAID)-induced enteropathy, and, rarely, stricture due to diverticulitis. Anastomotic strictures of the colon following a colorectal anastomosis occur in up to 30% of patients. Among patients who have developed stricture following a colorectal anastomosis, as many as 54% were persistently symptomatic [136]. Technical success of endoscopic balloon dilation in the treatment of anastomotic and benign inflammatory strictures has been reported in >73% to 100% of patients with good long-term clinical efficacy, although repeated dilations are frequently needed [137]. Technical failure or the need for repeat dilation is associated with the inability to endoscopically reach the stricture, severe bowel angulation, and long strictures (>2 cm). Complications including perforation and bleeding after endoscopic balloon dilation occur in 2% of cases [137]. Surgery is required for strictures where endoscopic treatment fails, or malignancy is suspected.

Approximately 8–34% of patients with colorectal cancer are accompanied by partial or complete LBO, and the mortality rate of emergency surgery due to LBO is reported to be as high as 30% [138,139]. Since the self-expandable metal stent (SEMS) was first used by Dohmoto et al. in 1990 for palliation and has been used for bridge to surgery (BTS) since 1994 [140,141], it has been increasingly used for the management of malignant LBO, not only for a palliative aim but also for preoperative treatment in surgical candidates [142]. Colorectal stenting is the preferred treatment for palliative purposes in the context of malignant LBO, and according to a meta-analysis, the initial technical and clinical success rates are reported to be 88–100% and 86.1%, respectively [143]. Colorectal stenting for palliative purposes was associated with a shorter hospital stay, lower intensive care unit admission rate, and shorter period to initiation of chemotherapy [144]. However, in a recent randomized clinical trial, the study was terminated early because of the high incidence of perforation associated with stent insertion in the group receiving chemotherapy [145]. A careful approach is required as more late complications may occur due to the direct effects of tumor shrinkage and tissue necrosis, as well as an increased survival time. Complications associated with colorectal stenting include reobstruction (3–29%), migration (1–10%), perforation (0–12%), torsion, fecal incontinence, and anal pain [4]. Colonic stenting for BTS showed a higher successful rate of primary anastomosis and the avoidance of stoma compared to emergency surgery [146]. Regarding left-sided malignant LBO, preoperative decompression with an SEMS compared to emergency surgery showed good short-term outcomes and a safe long-term prognosis [147,148,149]. Based on these results, the updated version of the guidelines from the European Society of Gastrointestinal Endoscopy in 2020 recommend BTS for left-sided malignant LBO [4]. In contrast, studies on preoperative stenting in right-sided malignant LBO are lacking. The procedure time is long, and the technical success rate is low. However, in a recent meta-analysis, BTS for right-sided malignant LBO confers preferable short-term outcomes, as well as left-sided postoperative complications (OR = 0.78; 95% CI, 0.66–0.92) and mortality (OR = 0.51; 95% CI, 0.28–0.92) [150]. Although the research results should gradually accumulate, in the case of malignant lesions in the right colon, it is expected that good results will be shown compared to emergency surgery if a stent is properly inserted before surgery.

### 2.5. Functional Bowel Disorders

Functional bowel disorder (FBD) is a term used to describe conditions characterized by chronic lower GI symptoms occurring in the absence of organic disease [151]. FBD can be diagnosed in patients who have been excluded from having organic disease. Abdominal pain, diarrhea, constipation, and abdominal discomfort are common symptoms in the general population. Since these symptoms are non-specific, colonoscopy can be considered to confirm the organic cause. However, it may be inappropriate to perform colonoscopy in all symptomatic patients. In a study of 767 patients with GI symptoms, colonoscopy showed a high diagnostic yield only for symptoms of bleeding or diarrhea. In this study, the diagnostic yields were 40% in patients with bleeding and 31.2% in patients with diarrhea without bleeding. On the other hand, among 362 patients with non-bleeding symptoms, only 8 patients (2.2%) had a serious colonic pathology, showing a low diagnostic yield [152]. Suleiman et al. reported that colonoscopy for the diagnosis of irritable bowel syndrome is not recommended in the early stages of diagnosis because of its cost-effectiveness [153]. Furthermore, the European Panel on Appropriateness of Gastrointestinal Endoscopy II (EPAGE II) reported a lack of evidence for colonoscopy in patients with FBDs without alarming symptoms. According to the EPAGE II criteria, colonoscopy is not considered in patients with isolated chronic abdominal pain [154]. Guidelines recommend performing colonoscopy to rule out organic causes before diagnosing FBD only in patients with alarming features (Table 4) [155]. Asghar et al. reported no diagnostic yield of colonoscopy in patients without alarming features [155]. In summary, in patients with FBD symptoms, colonoscopy is needed to rule out organic diseases, especially those with alarming features.

### 2.6. Intestinal Bleeding

Colonoscopy is important in the diagnosis and treatment of lower GI bleeding. Common causes of acute lower GI bleeding include diverticulosis, ischemic colitis, colorectal polyps or neoplasms, angioectasias, post-polypectomy or surgical bleeding, IBD, infectious colitis, anal diseases (hemorrhoid and anal fissure), solitary rectal ulcer, rectal prolapse, colorectal varices, radiation proctopathy, NSAID-induced enterocolopathy, and Dieulafoy lesions [156,157]. Common causes also vary by age. Table 5 summarizes the causes of lower GI bleeding according to age [158,159].

Colonoscopy, CT angiography, fluoroscopic angiography, and radionuclide scanning are used to diagnose lower GI bleeding. A meta-analysis reported that the diagnostic sensitivity of CT angiography in acute lower GI bleeding was 85.2%, and the specificity was 92.1% [160]. Fluoroscopic angiography can localize the bleeding source in 25% to 70% of lower GI bleeding cases [161,162], and subsequent therapeutic intervention is possible once the bleeding source is identified. Red blood cell (RBC) scanning is a sensitive test that can detect bleeding, even with a small amount of bleeding of 0.05–0.1 mL/min. Yet, the detection of the bleeding source is possible at bleeding rates of 0.5 mL/min by angiography and 0.3–0.5 mL/min by CT angiography [163,164]. However, Feuerstein et al. reported that CT angiography was able to localize the bleeding site more accurately than RBC scintigraphy (53% vs. 30%, *p* = 0.008) [165].

Among CT angiography, fluoroscopic angiography, and RBC scanning, only angiography can identify and treat bleeding points simultaneously. However, angiography, CT, and radionuclide scanning require active bleeding to make a diagnosis, and there is a risk of radiation exposure. In addition, CT angiography and fluoroscopic angiography may cause contrast agent toxicity, and since fluoroscopic angiography is an invasive test, complications such as bleeding and infection at the catheter site may occur [166,167,168].

Colonoscopy can confirm the bleeding point and perform the subsequent endoscopic hemostasis simultaneously. Unlike other diagnostic tools, colonoscopy can assess the macroscopic findings of lesions, tissue samples can be taken through colonoscopic biopsy, and diagnosis can be made without active bleeding. Colonoscopy does not use a contrast agent and there is no radiation exposure. Therefore, colonoscopy is currently recommended as the first-line procedure for acute lower GI bleeding in the guidelines. However, in hemodynamically unstable conditions, immediate colonoscopy is not recommended. Instead, hemodynamic resuscitation and other diagnostic modalities such as CT angiography or angiography are recommended. If upper GI bleeding is suspected, an upper GI endoscopy should be performed before a colonoscopy [157,169,170,171].

Endoscopic treatment of lower GI bleeding follows the principles of general endoscopic hemostasis. There are injections (e.g., epinephrine injection), thermal devices (e.g., electrosurgical unit), and mechanical therapies (e.g., endoclipping and endoscopic band ligation) for endoscopic hemostasis [156]. The best hemostasis method should be selected according to the endoscopist’s experience and the type of lesion, and if the first method fails, a second attempt can be made with another method [167]. However, epinephrine injection can be selected as the first method of hemostasis because of its high initial hemostasis rate, but it should be used together with other hemostasis techniques, as its rebleeding rate is high, at 6–36% [156,172]. Lower GI bleeding that usually requires endoscopic hemostasis includes diverticular bleeding, angioectasia, or post-polypectomy bleeding [173,174]. Diverticular bleeding accounts for 26% to 40% of lower GI bleeding cases [175]. Colonic diverticulum is caused by herniation of the mucosa and submucosa without a muscular layer at the point where vasa recta penetration occurs. Diverticular bleeding occurs when the vasa recta of the diverticulum is ruptured [176]. Diverticular bleeding is spontaneously resolved in 75–80% of cases, and early rebleeding after endoscopic hemostasis is uncommon [173]. However, the late rebleeding rate is high, ranging from 0% to 40% [175]. Epinephrine injection, thermal devices, and mechanical therapy can be used for endoscopic hemostasis of diverticular bleeding, but epinephrine involves a risk of rebleeding; complications such as perforation due to thermal injury must also be considered when using thermal devices. Mechanical therapies have the advantage of causing less tissue damage than thermal devices. In mechanical therapy, techniques such as direct endoclipping, indirect endoclipping, and endoscopic band ligation are generally used. Direct clipping captures the blood vessels in the diverticulum directly with a clip, whereas indirect clipping uses a clip to close the diverticulum like a zipper. Kishino et al. reported that direct clipping was associated with a reduced risk of early rebleeding compared to indirect clipping in a large multicenter cohort study [177]. It was also reported that no additional bleeding occurred when endoscopic band ligation was performed in a patient with early rebleeding after direct clipping [178]. According to an article by Lisa et al., when reviewing 137 cases, early rebleeding occurred in 0% of cases of endoclipping and banding, which was lower than that in other treatments (thermal contact, 12%; epinephrine, 15%; thermal contact plus injection, 24%). On the other hand, the late rebleeding rate was highest for endoclipping, at 17% [167]. Usually, diverticular bleeding is diagnosed presumptively when a colonic diverticulum is present in the absence of an obvious cause of lower GI bleeding [173]. The stigmata of recent hemorrhage in diverticular bleeding include active bleeding, non-bleeding visible vessels, adherent clots, and active bleeding after removal of a clot [179]. Therefore, it is important to closely observe the diverticula during colonoscopy, even if there is no overt bleeding.

Angioectasia, one of the main causes of lower GI bleeding, most often occurs in the small bowel or right colon, and asymptomatic angioectasia is not treated [180]. Patients receiving radiation therapy for prostate cancer or gynecological cancer may develop rectal angioectasia due to radiation proctitis. Radiation colitis occurs 9 months to 4 years after receiving radiation therapy [181,182,183]. If a history of radiation therapy is confirmed through history taking of patients with bleeding, the cause of bleeding can be easily estimated as angioectasia. Two-thirds of patients with angioectasia bleeding are aged >70 years, and the risk of bleeding increases with the use of anticoagulants or antiplatelet drugs and the accompanying underlying disease [171,184,185,186]. Thermal devices, especially argon plasma coagulation, are commonly used to treat angioectasia bleeding [170]. Active bleeding of angioectasia is well controlled with endoscopic treatment, but new lesions of angioectasia may develop elsewhere, and tend to rebleed. Saperas et al. reported that endoscopic argon plasma coagulation therapy was not associated with a reduction in recurrent bleeding [187]. A systemic review also reported that there was no difference in the rebleeding rate between the endoscopic and conservative treatment of angioectasia [188].

The frequency of bleeding after polypectomy varies in the literature. Numerous large-scale studies have reported that the frequency of bleeding after polypectomy ranges from 0.1% to 0.6% [189]. One study involving 53,220 colonoscopy cases reported a bleeding rate of 8.7/1000 procedures [190]. Risk factors associated with polypectomy include a polyp size > 10 mm, use of anticoagulants and antiplatelet drugs, pedunculated polyp with a thick stalk, location (right colon), age ≥ 65 years, cardiovascular or chronic renal disease, and polyp pathology [191]. Most bleeding after polypectomy is immediate bleeding, and bleeding can be managed with immediate endoscopic treatment. Endoscopic hemostasis is commonly used even in cases of delayed bleeding, and epinephrine injection, thermal devices, and mechanical therapy can be used [171]. Still, it is recommended to use endoclipping to minimize tissue damage [170].

## 3. Endocytoscopy

Recently, endocytoscopy has been introduced for the diagnosis of colorectal lesions, especially colorectal cancer. Previously, magnifying endoscopy and electronic chromoendoscopy (e.g., NBI) were used for the endoscopic diagnosis of colorectal cancer. Endocytoscopy is a high-magnification endoscopic system that has higher resolution than a magnifying endoscopy, and is capable of “optical biopsy”, which can visualize cell-level images [192,193]. Endocytoscopy can accurately evaluate mucosa and differentiate between normal and abnormal mucosa in real time [194]. In 2004, Inoue et al. first reported the observation of living cells in the esophagus, stomach, and colon using a catheter-type prototype endocytoscopy system (Olympus Optical, Co., Tokyo, Japan) [193]. Since then, endocytoscopy has been widely used for the diagnosis of colonic lesions and has shown accurate diagnostic performance comparable to pathological diagnosis. In 2011, Kudo et al. presented a new endocytoscopic classification (the EC classification) for colorectal lesions in a pilot study. Table 6 shows the EC classification presented by Kudo et al. [195]. EC classification has three tiers: EC1, EC2, and EC3. EC1 corresponds to non-neoplasia, EC2 to dysplasia, and EC3 to cancerous lesions. EC1 is classified into normal mucosal EC1a and hyperplastic polyp EC1b. EC3 is classified as EC3a corresponding to high-grade dysplasias or submucosal cancer, and EC3b corresponds to submucosal cancer or higher. Kudo et al. analyzed 213 samples using this classification, and were able to distinguish between non-neoplastic and neoplastic lesions with 100% sensitivity and 100% specificity (*p* < 0.05). In addition, they showed a sensitivity of 90.1% and specificity of 99.2% (*p* < 0.05) in distinguishing massively invasive submucosal cancer from other neoplastic lesions [195].

Kudo et al. proposed a new EC-V pattern combining endocytoscopy and NBI, and compared it with magnifying endoscopy with NBI, pit pattern, and EC-C pattern. Compared with the EC-C pattern, the EC-V pattern was slightly less accurate in predicting invasive cancer (*p* = 0.04), but was comparable to NBI and pit pattern. Diagnosis using the EC-V pattern took a shorter examination time than using the EC-C pattern (*p* < 0.001) [196]. Although the EC method requires the use of methylene blue, the EC-V method has the advantage of a shorter procedure time and cost-efficiency, as microvascular irregularity can be evaluated without staining [196].

Several studies applying endocytoscopy in the management of IBD have also been published. Ueda et al. classified EC appearance into four categories, EC-A, EC-B, EC-C, and EC-D, and showed correlations with MES, clinical activity, and pathological microscopic features of UC [197]. Kazumi et al. evaluated goblet cells in 120 patients with an MES of 0 using endocytoscopy, and found that depleted goblet cells had a higher clinical relapse rate than non-depleted goblet cells (*p* = 0.02) [198]. Maeda et al. evaluated 52 patients with UC by endocytoscopy NBI, and the sensitivity, specificity, positive predictive value, negative predictive value, and accuracy of EC-NBI in diagnosing acute inflammation were 84.0%, 100%, 87.1%, 100% and 92.3%, respectively. Compared to conventional endoscopy, the diagnostic specificity, negative predictive value, and accuracy were significantly superior [199]. In a prospective study involving 40 patients with IBD (CD, n = 19; UC, n = 21), Neumann et al. reported that the concordance between endocytoscopy and histopathology was 100% in evaluating disease activity [200].

## 4. Artificial Intelligence and Magnetically Controlled Capsule Endoscopy

The complete resection of neoplastic lesions (such as adenomas) during colonoscopy is considered a reliable measure to reduce both the incidence and mortality of CRC [201]. Poorly conducted colonoscopy may lead to missed lesions and impair CRC prevention [202]. To improve the effectiveness of colonoscopy screening, achieving high ADR is necessary [203]. The use of artificial intelligence (AI), specifically deep neural networks (e.g., so-called deep learning), has emerged as a promising tool to address challenges in colonoscopy. By utilizing computer-aided polyp detection (CADe) and classification (CADx) in real-time during colonoscopy, AI has enabled endoscopists to improve their ADR and interpret polyp histology with greater accuracy [204]. Prospective studies have shown promising results for both CADe and CADx, and retrospective benchmark tests are currently being conducted for further validation [205,206]. Many healthcare corporations, including endoscopy manufacturers, have launched AI products for colonoscopy after conducting dedicated testing in collaboration with academic partners. These products, such as ENDO-AID (Olympus Corp., Tokyo, Japan), CAD EYE (Fujifilm Corp., Tokyo, Japan), Discovery (Pentax Corp., Tokyo, Japan), GI Genius (Medtronic Corp., Dublin, Ireland), and EndoBRAIN (Cybernet Corp., Tokyo, Japan), have obtained regulatory approval in Europe and Japan within a relatively short period (2018–2020). The availability of these AI products on the market raises the possibility of the more widespread adoption of AI in colonoscopy. There have been six randomized controlled trials (RCTs) published that have investigated the potential of CADe in colonoscopy, providing strong evidence in the field [207,208,209,210,211,212]. Representative studies have shown that Repici et al. utilized the CADe system GI Genius (Medtronic Corp., Dublin, diagnostic sensitivity 99.7%), which detects and visualizes colorectal polyps in real-time, to confirm the ADR in 685 high-resolution colonoscopy examinations conducted at multiple institutions. They found that utilizing AI significantly increased the ADR compared to the control group (54.8% vs. 40.4%, hazard ratio 1.3) [207]. Wang et al. evaluated more than 520 colonoscopies, and found that the adenoma detection rate was significantly increased with the use of a CADe system compared to the control group (29.1% vs. 20.3%) [208]. Subsequently, two meta-analyses were conducted to assess the impact of these RCTs [205,213]. Both meta-analyses reported that the use of CADe during colonoscopy is likely to increase the adenoma detection rate by approximately 50%, which could have significant benefits in cancer prevention. Furthermore, the trials found no major drawbacks to the use of AI, including serious adverse events. Based on these positive findings, the ESGE has recently published a guideline that weakly recommends the adoption of AI during colonoscopy [214]. In contrast, there have been no randomized controlled trials (RCTs) conducted in the CADx area. However, several large prospective studies have been published, two of which have successfully demonstrated the validity of using AI to implement a diagnose-and-leave strategy [215,216]. A study conducted in Japan used the AI model EndoBRAIN (Cybernet Corp., Tokyo) to classify 100 lesions that were 10 mm or smaller as non-neoplastic or neoplastic using around 69,000 chromoendoscopy and narrow-band imaging (NBI) images. The results showed significantly higher accuracy (96–98%) than that of endoscopists (93.3–94.6%) and trainees (69–70.4%) [217].

AI and computer-assisted diagnosis are also used in the treatment of IBD. Bossuyte et al. trained an algorithm to integrate color data of pixels and the red channel of red–green–blue pixel values with blood vessel pattern recognition. The score generated by this system (Red Density) showed a strong correlation with the MES (r = 0.76), UCEIS (r = 0.74), and histologic score (Robarts Histopathology Index, r = 0.74) [218]. Takenaka et al. conducted a prospective study using a convolution neural network on 875 patients with UC, and reported that endoscopic remission was identified with 90.1% accuracy and histological remission with 92.9% accuracy [219]. In addition, Mossoto et al. classified the disease by applying machine learning to endoscopic and histologic data of 287 patients diagnosed with pediatric IBD, and reported that 83.3% of pediatric patients with IBD were accurately classified [220], demonstrating the usefulness of AI in diagnosis. Moreover, a study that developed CAD using endocytoscopy was also published. Maeda and Kudo et al. obtained endocytoscopic images and biopsy samples for each segment of UC, and used them for machine learning. Their CAD reported that the diagnostic sensitivity, specificity, and accuracy were 74%, 97%, and 91%, respectively, and reproducibility was k = 1 [221]. Endoscopic AI is developing at a remarkably rapid pace, and several products are already being used in clinical settings. Although it is still controversial whether AI can follow the diagnostic performance of experienced endoscopists, it is thought that it can be used for various types of intestinal diseases through continuous development.

Since the introduction of the first capsule endoscopy (CE) in 2001, it has become a preferred means of examination for the small intestine due to its non-invasiveness, accuracy, and patient comfort [222]. Recently, the diagnostic applications of CE have expanded to upper and lower gastrointestinal disorders with the invention of esophageal CE [223] and colon CE [224]. Additionally, magnetically controlled capsule endoscopy (MCE) was developed to achieve complete visualization of the stomach [225]. MCE uses a capsule containing a miniature camera that is propelled through the gastrointestinal tract using magnetic fields, and is controlled from outside the body using a magnetic controller. This allows for greater control of the capsule’s movement and the ability to obtain detailed images of the entire gastrointestinal tract. MCE has the potential to improve diagnostic accuracy and patient comfort compared to traditional endoscopic procedures. Compared with traditional small bowel CE, which is usually performed after negative findings on gastroscopy and colonoscopy, MCE can examine the stomach and the small bowel at the same time, simplifying the clinical examination process. Previous studies have demonstrated that MCE has comparable diagnostic accuracy to conventional endoscopy, and is widely used in clinical practice [226,227]. The favorable application of MCE to different parts of the gastrointestinal tract may imply its potential to replace traditional endoscopy in certain scenarios. However, further studies are required to achieve one-time overall gastrointestinal tract examinations.

## 5. Conclusions

Colonoscopy is the most commonly performed endoscopic procedure. Based on a recent review of the literature, several key conclusions are highlighted in this review. For infectious diseases, colonoscopy is helpful for the differential diagnosis in revealing endoscopic gross findings and obtaining the specimens for pathology. Additionally, colonoscopy aids in the post-treatment monitoring of IBD and provides clues for distinguishing between infectious disease and IBD. Colonoscopy is essential for the diagnosis of neoplasms that are diagnosed only through pathological confirmation. Recently, early colorectal cancers are being commonly treated using colonoscopy. Moreover, the characteristics of tumors can be described in more detail by image-enhanced endoscopy and magnifying endoscopy. Colonoscopy can be helpful for the endoscopic decompression of colonic volvulus in LBO, balloon dilatation as a treatment for benign stricture, and colon stenting as a treatment for malignant obstruction. For the diagnosis of functional bowel disorder, colonoscopy is used to investigate other organic causes of symptom. In the field of endoscopy, research on the use of AI is being actively conducted. Finally, with the introduction of a computer-aided diagnostic system through deep learning, AI can discriminate various images more quickly and objectively than humans.

## Figures and Tables

**Table 1 diagnostics-13-01262-t001:** Prevalent sites of infectious enterocolitis according to the causative microorganism.

Prevalent Site of Infection	Causative Microorganism
Distal small bowel	*Yersinia* *Salmonella* *Shigella* *Campylobacter*
Distal ileum and cecum	Tuberculosis Amoebiasis
Right colon	*Salmonella* Amoebiasis *Yersinia*
Left colon	*Shigella* *Gonorrhea* *Chlamydia*
Pancolitis	*Escherichia coli* *Clostridium difficile* Cytomegalovirus

**Table 2 diagnostics-13-01262-t002:** Endoscopic scoring systems for IBD.

Scoring System	Disease Type	Criteria
MES	UC	0: Normal or inactive disease 1: Mild disease (erythema, decreased vascular pattern, mild friability) 2: Moderate disease (marked erythema, absent vascular pattern, friability, erosions) 3: Severe disease (spontaneous bleeding, ulceration)
UCEIS		Combines vascular pattern, bleeding, erosions and ulcers, and evaluates the severity on a scale of 0 to 8
UCCIS		Evaluates 4 parameters: granularity, vascular pattern, ulceration, and bleeding/friability Score range: 0–12, with higher scores indicating more severe disease
CDEIS	CD	Considers the surface affected by disease, ulcerations, and ulcerated surface Score range: 0–44, with higher scores indicating more severe disease
SES-CD		Evaluates 4 parameters: size of ulcers, ulcerated surface, affected surface, and presence of narrowing Score range: 0–56, with higher scores indicating more severe disease

MES, Mayo endoscopic score; UCEIS, Ulcerative Colitis Endoscopic Index of Severity; UCCIS, Ulcerative Colitis Colonoscopic Index of Severity; CDEIS, Crohn’s Disease Endoscopic Index of Severity; SES-CD, Simple Endoscopic Score for Crohn’s Disease.

**Table 3 diagnostics-13-01262-t003:** Characteristic features of colorectal subepithelial tumors.

Type of Lesion	Layer of Origin	EUS Appearance
Benign lesions		
Lipoma	Third	Hyperechoic, homogenous, smooth margin
Lymphangioma	Second, Third	Anechoic with internal septa, serpiginous shape
Leiomyoma	Second, Fourth	Hypoechoic (similar to the muscular layer), homogenous, round or oval, well-circumscribed
Granular cell tumor	Second, Third	Hypoechoic (higher echogenicity compared to the muscular layer), heterogenous, smooth margin
Schwannoma	Third, Fourth	Hypoechoic, homogenous, smooth margin, sometimes with marginal halo
Calcifying fibrous tumor	Second, Third, Forth	Hypoechoic, post-acoustic shadowing with slightly hyperechoic foci inside
Rectal tonsil	Second, Third	Hypoechoic, well-demarcated
Endometriosis	Forth, Fifth	Hypoechoic. Heterogenous (mighht extended into the rectovaginal setum), irregular margin
Lesions with malignant potential		
Neuroendocrine tumor	Second, Third	Hypoechoic or isoechoic, homogenous, smooth margin
GIST—low risk	Second, Fourth	Hypoechoic, round, <3 cm, heterogenous, round, smooth margin
GIST—high risk	Second, Fourth	Hypoechoic, >3 cm, heterogenous with cystic spaces or echogenic foci, irregular margin
MALToma	Second, Third	Hypoechoic, Partial indentation of the submucosa layer

GIST, Gastrointestinal stromal tumor; MALToma, mucosa-associated lymphoid tissue lymphoma; EUS, endoscopic ultrasonography.

**Table 4 diagnostics-13-01262-t004:** Lower gastrointestinal alarm features that require colonoscopy to rule out organic causes [155].

Symptom onset ≥ 45 y
Nocturnal bowel symptoms
Unintentional weight loss
Recent change in bowel habit
Rectal bleeding without documented bleeding hemorrhoids or anal fissures
Family history of inflammatory bowel disease or colorectal cancer
Evidence of inflammation on blood or stool testing
Evidence of iron deficiency anemia
Abnormal gastrointestinal examination

**Table 5 diagnostics-13-01262-t005:** Common causes of lower gastrointestinal bleeding according to age [158,159].

Children and Adolescents	Adults	Elderly People (>60 y)
Anal fissure Meckel diverticulum Juvenile polyps Inflammatory bowel diseases	Diverticular disease Inflammatory bowel disease Neoplasms Infectious colitis	Diverticular disease Angiodysplasia Neoplasms Ischemic colitis

**Table 6 diagnostics-13-01262-t006:** Endocytoscopic classification.

Classification	Endocytoscopic Findings	Histopathology
EC1a	Fusiform nuclei and roundish lumens	Non-neoplasia
EC1b	Small roundish nuclei and serrated lumens	
EC2	Fusiform or roundish nuclei and slit-like lumens	Dysplasia
EC3a	A large number of roundish nuclei and irregular lumens	High-grade dysplasia or slightly invasive submucosal cancer
EC3b	Distorted nuclei and unclear gland formation	Massively invasive submucosal cancer or worse

## Data Availability

Not applicable.

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
