# Peer review of "A Review of Colonoscopy in Intestinal Diseases"

_diagnostics, 2023, doi:10.3390/diagnostics13071262_

Round 1

Reviewer 1 Report

·       It seems to me that the issue is important and necessary. the review in general is well organized. however, I think that the item infections and colonoscopy is very extensive and does not provide so many elements of practical use. in that sense I recommend shortening it and, if possible, including images that accompany what is described.

·       In the rest of the text, the quality (aesthetic aspect) of the tables should be improved.

·       Another chapter that i think should be enriched is that referring to neoplasms. It seems important to me to perform an analysis of the literature in differences between the impact of early diagnosis and mortality of screening programs with rectosigmoidoscopy and complete colonoscopy. At this point the two techniques should be compared. the recommendation for preventive colonoscopies for cancer should be updated (for example, indicate guidelines such as task force or others). It would also be important to make a critical comment on the new england publication of the nordic group regarding the impact of colonoscopies on colonic cancer mortality. Finally, the quality criteria in colonoscopy necessary to be able to follow the recommendations of colonoscopic follow-up (intestinal cleansing and its scales, withdrawal time, arrival at the cecum etc ...) should be correctly described.

Author Response

Response to Reviewer Comments

Point 1: I recommend shortening it and, if possible, including images that accompany what is described.

Response 1

Thank you for your kind comments and thoughtful suggestions. Sentences have been abbreviated and the content has been reduced for the infection and colonoscopy sections

Point 2: In the rest of the text, the quality (aesthetic aspect) of the tables should be improved.

Response 2

Thank you for your kind comments and thoughtful suggestions. I have corrected the quality (aesthetic aspect) of the tables.

Point 3: Another chapter that i think should be enriched is that referring to neoplasms. It seems important to me to perform an analysis of the literature in differences between the impact of early diagnosis and mortality of screening programs with rectosigmoidoscopy and complete colonoscopy. It would also be important to make a critical comment on the new england publication of the nordic group regarding the impact of colonoscopies on colonic cancer mortality.

Response 3

Thank you for your kind comments and thoughtful suggestions. I have performed a literature analysis on the difference between the impact of early diagnosis and the mortality of screening programs using recto-sigmoidoscopy and colonoscopy in the neoplastic part. The following paragraph has been inserted. Additionally, I have provided critical comments on the New England publication by the Nordic group regarding the impact of colonoscopies on colonic cancer mortality.

Colorectal cancer (CRC) is a major cause of morbidity and mortality throughout the world. It accounts for over 10% of all cancer incidence [94]. It is the third most common cancer worldwide and the second most common cause of death [94]. Most guidelines, including those from the American Cancer Society [95], the US Preventive Services Task Force [96], and the European Society of Gastrointestinal Endoscopy (ESGE) [97], recommend screening for CRC in average-risk individuals beginning at the age of 45 or 50 years. Both colonoscopy and sigmoidoscopy can detect and remove polyps, potentially preventing malignant transformation and decreasing CRC mortal-ity and incidence. To date, four large randomized controlled trials comparing flexible sigmoidoscopy screening with no screening showed reductions in CRC incidence (18%-23%) and CRC mortality (22%-33%) [98-101]. These findings provide substantial protection against CRC diagnosis and death, and the benefits can last for up to 17 years [102]. Randomized controlled trials of screening colonoscopy are ongoing, but definitive results will not be available until 2022 or 2026–2027 [103-105]. Cohort and case–control studies found an association between lower endoscopy and reduced CRC mortality and incidence. A large prospective cohort study of nearly 89,000 nurses and other health care professionals found that, over 24 years of follow up, colonoscopy was associated with a 68% reduction (95% CI, 0.55–0.76) in CRC-specific mortality com-pared with no exposure to colonoscopy [106]. Individuals who underwent colonoscopy with polypectomy were found to have a 43% reduction in CRC incidence compared to those with no lower endoscopy [106]. However, cohort study probably overestimates the real-world effectiveness of colonoscopy because of the inability to adjust for im-portant factors such as incomplete adherence to testing and the tendency of healthier persons to seek preventive care. In a Canadian case-control study, any colonoscopy was associated with a 37% reduction in the odds of CRC death [107]. Similar case-control studies using the Surveillance, Epidemiology, and End Results (SEER)-Medicare and Veterans Administration data also found approximately 60% reductions in CRC death associated with colonoscopy, with similar differences by site [108,109]. However, these three case-control studies were unable to determine indica-tions for colonoscopy and excluded colonoscopies performed within 6 months of CRC diagnosis, likely introducing bias. In a meta-analysis conducted with 13 cohorts in-cluding 4,713,778 individuals and 16 case-control studies, colonoscopy screening not only reduced the incidence of colorectal cancer by 52% (risk ratio [RR]: 0.48, 95% CI, 0.46–0.49), but also reduced colorectal cancer related mortality by 62% (RR: 0.38, 95% CI, 0.36–0.40) [110]. Flexible sigmoidoscopy and colonoscopy are both recommended CRC screening strategies, but their relative effectiveness is unclear. According to the case-control study using the SEER-Medicare database, screening colonoscopy was as-sociated with a greater reduction of 74% (OR 0.26, 95% CI, 0.23–0.30) in CRC mortality compared to screening sigmoidoscopy, which was associated with a 35% reduction (OR 0.65, 95% CI, 0.48–0.89) in CRC mortality. Additionally, screening colonoscopy was found to be more effective in reducing mortality in the distal colon compared to the proximal colon.

Point 4: Finally, the quality criteria in colonoscopy necessary to be able to follow the recommendations of colonoscopic follow-up (intestinal cleansing and its scales, withdrawal time, arrival at the cecum etc ...) should be correctly described.

Response 4

Thank you for your kind comments and thoughtful suggestions. I have described the quality criteria necessary for colonoscopy in order to be able to follow the recommendations for colonoscopic follow-up. The following paragraph has been inserted.

Improving colonoscopy screening results is crucial for the early detection and prevention of colorectal cancer [111]. Recording quality indicators is essential for as-sessing the effectiveness of population-based colonoscopy screening programs. The quality indicators vary between countries, such as the United States [111] and the United Kingdom [112], but they generally include the following:

  1. Consent obtained: Ensuring informed consent is obtained from patients before the procedure.
  2. Cecal insertion rate: A high rate (97% or higher in the US, 90% minimum in the UK) indicates successful navigation of the colonoscope to the cecum.
  3. Adequate bowel preparation: A clean colon is necessary for accurate visuali-zation; the suggested rates are 85% or higher in the US and 90-95% in the UK.
  4. Adenoma detection rate (ADR): The percentage of patients with at least one adenoma detected during colonoscopy; higher rates (25% or more in the US, 35-40% in the UK) indicate better screening quality.
  5. Withdrawal time: Time taken for the colonoscope to be withdrawn after reaching the cecum; longer times (6 minutes or more in the US, 6-10 minutes in the UK) are associated with improved adenoma detection.
  6. Complication rates: Low rates of complications, such as perforation (1/1,000 or less) and bleeding after polypectomy (1% or less in the US, 1/100 or less in the UK).
  7. Polyp retrieval rate: The percentage of removed polyps that are successfully retrieved for histopathological examination (90-95% in the UK).

The NordICC (Nordic-European Initiative on Colorectal Cancer) study highlights the importance of quality control in population-based colonoscopy screening programs. A significant issue identified in this study is the low quality of colonoscopy screenings, which can affect the effectiveness of these programs in detecting and preventing CRC [113]. The ongoing NordICC study aims to evaluate the long-term performance of co-lonoscopy screening and the impact of quality control measures. In the next 5 years, the study is expected to reveal valuable insights into the effectiveness of various qual-ity indicators in improving colonoscopy screening results [104]. By examining these results, healthcare professionals and policymakers can make informed decisions about implementing and refining population-based colonoscopy screening programs.

Reviewer 2 Report

The manuscript is well written and provides a comprehensive overview of the use of colonoscopy in intestinal disease. However, I would recommend adding a section on the future of colonoscopy, especially regarding the use of artificial intelligence (AI) and its impact on polyp detection rates. It would also be useful to discuss how providers perceive the integration of AI in colonoscopy.

Additionally, it would be beneficial to briefly comment on other modalities that are being developed, such as magnetically-controlled capsule endoscopy

Author Response

Response to Reviewer Comments

Point 1: I would recommend adding a section on the future of colonoscopy, especially regarding the use of artificial intelligence (AI) and its impact on polyp detection rates. It would also be useful to discuss how providers perceive the integration of AI in colonoscopy.

Response 1

Thank you for your kind comments and thoughtful suggestions. I have separated the section on AI and inserted the following paragraph.

Complete resection of neoplastic lesions (such as adenomas) during colonoscopy is considered a reliable measure to reduce both the incidence and mortality of CRC [201]. Poorly conducted colonoscopy may lead to missed lesions and impair CRC prevention [202]. To improve the effectiveness of colonoscopy screening, achieving high ADR is necessary [203]. The use of Artificial intelligence (AI), specifically deep neural net-works (e.g. so‐called deep learning), has emerged as a promising tool to address chal-lenges in colonoscopy. By utilizing computer-aided polyp detection (CADe) and classi-fication (CADx) in real-time during colonoscopy, AI has enabled endoscopists to im-prove their ADR and interpret polyp histology with greater accuracy [204]. Prospec-tive studies have shown promising results for both CADe and CADx, and retrospective benchmark tests are currently being conducted for further validation [205,206]. Many healthcare corporations, including endoscopy manufacturers, have launched AI prod-ucts for colonoscopy after conducting dedicated testing in collaboration with academic partners. These products, such as ENDO-AID (Olympus Corp., Tokyo), CAD EYE (Fuji-film Corp., Tokyo), Discovery (Pentax Corp., Tokyo), GI Genius (Medtronic Corp., Dublin), and EndoBRAIN (Cybernet Corp., Tokyo), have obtained regulatory approval in Europe and Japan within a relatively short period (2018–2020). The availability of these AI products on the market raises the possibility of more widespread adoption of AI in colonoscopy. There have been six randomized controlled trials (RCTs) published that have investigated the potential of CADe in colonoscopy, providing strong evi-dence in the field [207-212]. Representative studies have shown that Repici et al. uti-lized the CADe system GI Genius (Medtronic Corp., Dublin, diagnostic sensitivity 99.7%), which detects and visualizes colorectal polyps in real-time, to confirm the ADR in 685 high-resolution colonoscopy examinations conducted at multiple institutions. They found that utilizing AI significantly increased the ADR compared to the control group (54.8% vs. 40.4%, hazard ratio 1.3) [207]. Wang et al. evaluated more than 520 colonoscopies and found that the adenoma detection rate was significantly increased with the use of a CADe system compared to the control group (29.1% vs. 20.3%) [208]. Subsequently, two meta-analyses were conducted to assess the impact of these RCTs [205,213]. Both meta-analyses reported that the use of CADe during colonoscopy is likely to increase the adenoma detection rate by approximately 50%, which could have significant benefits in cancer prevention. Furthermore, the trials found no major drawbacks to the use of AI, including serious adverse events. Based on these positive findings, the ESGE has recently published a guideline that weakly recommends the adoption of AI during colonoscopy [214]. In contrast, there have been no randomized controlled trials (RCTs) conducted in the CADx area. However, several large prospec-tive studies have been published, two of which have successfully demonstrated the va-lidity of using AI to implement a diagnose-and-leave strategy [215,216]. A study con-ducted in Japan used the AI model EndoBRAIN (Cybernet Corp., Tokyo) to classify 100 lesions that were 10mm or smaller as non-neoplastic or neoplastic using around 69,000 chromoendoscopy and narrow-band imaging (NBI) images. The results showed signif-icantly higher accuracy (96-98%) than that of endoscopists (93.3-94.6%) and trainees (69-70.4%) [217].

Ref.

  1. Corley, D.A.; Levin, T.R.; Doubeni, C.A. Adenoma detection rate and risk of colorectal cancer and death. N Engl J Med 2014, 370, 2541, doi:10.1056/NEJMc1405329.
  2. Kaminski, M.F.; Regula, J.; Kraszewska, E.; Polkowski, M.; Wojciechowska, U.; Didkowska, J.; Zwierko, M.; Rupinski, M.; Nowacki, M.P.; Butruk, E. Quality indicators for colonoscopy and the risk of interval cancer. N Engl J Med 2010, 362, 1795-1803, doi:10.1056/NEJMoa0907667.
  3. Vinsard, D.G.; Mori, Y.; Misawa, M.; Kudo, S.E.; Rastogi, A.; Bagci, U.; Rex, D.K.; Wallace, M.B. Quality assurance of computer-aided detection and diagnosis in colonoscopy. Gastrointest Endosc 2019, 90, 55-63, doi:10.1016/j.gie.2019.03.019.
  4. Barua, I.; Vinsard, D.G.; Jodal, H.C.; Loberg, M.; Kalager, M.; Holme, O.; Misawa, M.; Bretthauer, M.; Mori, Y. Artificial intelligence for polyp detection during colonoscopy: a systematic review and meta-analysis. Endoscopy 2021, 53, 277-284, doi:10.1055/a-1201-7165.
  5. Kudo, S.E.; Mori, Y.; Misawa, M.; Takeda, K.; Kudo, T.; Itoh, H.; Oda, M.; Mori, K. Artificial intelligence and colonoscopy: Current status and future perspectives. Dig Endosc 2019, 31, 363-371, doi:10.1111/den.13340.
  6. Repici, A.; Badalamenti, M.; Maselli, R.; Correale, L.; Radaelli, F.; Rondonotti, E.; Ferrara, E.; Spadaccini, M.; Alkandari, A.; Fugazza, A.; et al. Efficacy of Real-Time Computer-Aided Detection of Colorectal Neoplasia in a Randomized Trial. Gastroenterology 2020, 159, 512-520 e517, doi:10.1053/j.gastro.2020.04.062.
  7. Wang, P.; Liu, X.; Berzin, T.M.; Glissen Brown, J.R.; Liu, P.; Zhou, C.; Lei, L.; Li, L.; Guo, Z.; Lei, S.; et al. Effect of a deep-learning computer-aided detection system on adenoma detection during colonoscopy (CADe-DB trial): a double-blind randomised study. Lancet Gastroenterol Hepatol 2020, 5, 343-351, doi:10.1016/S2468-1253(19)30411-X.
  8. Wang, P.; Berzin, T.M.; Glissen Brown, J.R.; Bharadwaj, S.; Becq, A.; Xiao, X.; Liu, P.; Li, L.; Song, Y.; Zhang, D.; et al. Real-time automatic detection system increases colonoscopic polyp and adenoma detection rates: a prospective randomised controlled study. Gut 2019, 68, 1813-1819, doi:10.1136/gutjnl-2018-317500.
  9. Su, J.R.; Li, Z.; Shao, X.J.; Ji, C.R.; Ji, R.; Zhou, R.C.; Li, G.C.; Liu, G.Q.; He, Y.S.; Zuo, X.L.; et al. Impact of a real-time automatic quality control system on colorectal polyp and adenoma detection: a prospective randomized controlled study (with videos). Gastrointest Endosc 2020, 91, 415-424 e414, doi:10.1016/j.gie.2019.08.026.
  10. Gong, D.; Wu, L.; Zhang, J.; Mu, G.; Shen, L.; Liu, J.; Wang, Z.; Zhou, W.; An, P.; Huang, X.; et al. Detection of colorectal adenomas with a real-time computer-aided system (ENDOANGEL): a randomised controlled study. Lancet Gastroenterol Hepatol 2020, 5, 352-361, doi:10.1016/S2468-1253(19)30413-3.
  11. Liu, W.N.; Zhang, Y.Y.; Bian, X.Q.; Wang, L.J.; Yang, Q.; Zhang, X.D.; Huang, J. Study on detection rate of polyps and adenomas in artificial-intelligence-aided colonoscopy. Saudi J Gastroenterol 2020, 26, 13-19, doi:10.4103/sjg.SJG_377_19.
  12. Hassan, C.; Spadaccini, M.; Iannone, A.; Maselli, R.; Jovani, M.; Chandrasekar, V.T.; Antonelli, G.; Yu, H.; Areia, M.; Dinis-Ribeiro, M.; et al. Performance of artificial intelligence in colonoscopy for adenoma and polyp detection: a systematic review and meta-analysis. Gastrointest Endosc 2021, 93, 77-85 e76, doi:10.1016/j.gie.2020.06.059.
  13. Bisschops, R.; East, J.E.; Hassan, C.; Hazewinkel, Y.; Kaminski, M.F.; Neumann, H.; Pellise, M.; Antonelli, G.; Bustamante Balen, M.; Coron, E.; et al. Advanced imaging for detection and differentiation of colorectal neoplasia: European Society of Gastrointestinal Endoscopy (ESGE) Guideline - Update 2019. Endoscopy 2019, 51, 1155-1179, doi:10.1055/a-1031-7657.
  14. Mori, Y.; Kudo, S.E.; Misawa, M.; Saito, Y.; Ikematsu, H.; Hotta, K.; Ohtsuka, K.; Urushibara, F.; Kataoka, S.; Ogawa, Y.; et al. Real-Time Use of Artificial Intelligence in Identification of Diminutive Polyps During Colonoscopy: A Prospective Study. Ann Intern Med 2018, 169, 357-366, doi:10.7326/M18-0249.
  15. Horiuchi, H.; Tamai, N.; Kamba, S.; Inomata, H.; Ohya, T.R.; Sumiyama, K. Real-time computer-aided diagnosis of diminutive rectosigmoid polyps using an auto-fluorescence imaging system and novel color intensity analysis software. Scand J Gastroenterol 2019, 54, 800-805, doi:10.1080/00365521.2019.1627407.
  16. Kudo, S.E.; Misawa, M.; Mori, Y.; Hotta, K.; Ohtsuka, K.; Ikematsu, H.; Saito, Y.; Takeda, K.; Nakamura, H.; Ichimasa, K.; et al. Artificial Intelligence-assisted System Improves Endoscopic Identification of Colorectal Neoplasms. Clin Gastroenterol Hepatol 2020, 18, 1874-1881 e1872, doi:10.1016/j.cgh.2019.09.009.

Point 2: It would be beneficial to briefly comment on other modalities that are being developed, such as magnetically-controlled capsule endoscopy.

Response 2

Thank you for your kind comments and thoughtful suggestions. I have inserted the following paragraph.

Since the introduction of the first capsule endoscopy (CE) in 2001, it has become a preferred examination for small intestine due to its non-invasiveness, accuracy, and patient comfort [222]. Recently, the diagnostic applications of CE have expanded to upper and lower gastrointestinal disorders with the invention of esophageal CE [223] and colon CE [224]. Additionally, magnetically-controlled capsule endoscopy (MCE) was developed to achieve complete visualization of the stomach [225]. MCE uses a capsule containing a miniature camera that is propelled through the gastrointestinal tract using magnetic fields, and is controlled from outside the body using a magnetic controller. This allows for greater control of the capsule's movement and the ability to obtain detailed images of the entire gastrointestinal tract. MCE has the potential to improve diagnostic accuracy and patient comfort compared to traditional endoscopic procedures. Compared with traditional small bowel CE, which is usually performed after negative findings on gastroscopy and colonoscopy, MCE can examine the stom-ach and the small bowel at one time, simplifying the clinical examination process. Pre-vious studies have demonstrated that MCE has comparable diagnostic accuracy to conventional endoscopy and is widely used in clinical practice [226,227]. The favorable application of MCE to different parts of the gastrointestinal tract may have the poten-tial to replace traditional endoscopy in certain scenarios. However, further studies are required to achieve one-time overall gastrointestinal tract examinations.

Ref.

  1. Eliakim, R. Video capsule endoscopy of the small bowel. Curr Opin Gastroenterol 2013, 29, 133-139, doi:10.1097/MOG.0b013e32835bdc03.
  2. Chen, Y.Z.; Pan, J.; Luo, Y.Y.; Jiang, X.; Zou, W.B.; Qian, Y.Y.; Zhou, W.; Liu, X.; Li, Z.S.; Liao, Z. Detachable string magnetically controlled capsule endoscopy for complete viewing of the esophagus and stomach. Endoscopy 2019, 51, 360-364, doi:10.1055/a-0856-6845.
  3. Enns, R.A.; Hookey, L.; Armstrong, D.; Bernstein, C.N.; Heitman, S.J.; Teshima, C.; Leontiadis, G.I.; Tse, F.; Sadowski, D. Clinical Practice Guidelines for the Use of Video Capsule Endoscopy. Gastroenterology 2017, 152, 497-514, doi:10.1053/j.gastro.2016.12.032.
  4. Carpi, F.; Galbiati, S.; Carpi, A. Magnetic shells for gastrointestinal endoscopic capsules as a means to control their motion. Biomed Pharmacother 2006, 60, 370-374, doi:10.1016/j.biopha.2006.07.001.
  5. Liao, Z.; Duan, X.D.; Xin, L.; Bo, L.M.; Wang, X.H.; Xiao, G.H.; Hu, L.H.; Zhuang, S.L.; Li, Z.S. Feasibility and safety of magnetic-controlled capsule endoscopy system in examination of human stomach: a pilot study in healthy volunteers. J Interv Gastroenterol 2012, 2, 155-160, doi:10.4161/jig.23751.
  6. Liao, Z.; Hou, X.; Lin-Hu, E.Q.; Sheng, J.Q.; Ge, Z.Z.; Jiang, B.; Hou, X.H.; Liu, J.Y.; Li, Z.; Huang, Q.Y.; et al. Accuracy of Magnetically Controlled Capsule Endoscopy, Compared With Conventional Gastroscopy, in Detection of Gastric Diseases. Clin Gastroenterol Hepatol 2016, 14, 1266-1273 e1261, doi:10.1016/j.cgh.2016.05.013.

Round 2

Reviewer 1 Report

I think this new redaction it's better